# Prevalence of Sarcopenia and Impact on Survival in Patients with Metastatic Gastroenteropancreatic Neuroendocrine Tumours

**DOI:** 10.3390/cancers15030782

**Published:** 2023-01-27

**Authors:** Dominique S. V. M. Clement, Monique E. van Leerdam, Soraya de Jong, Martin O. Weickert, John K. Ramage, Margot E. T. Tesselaar, Rajaventhan Srirajaskanthan

**Affiliations:** 1Kings Health Partners, ENETS Centre of Excellence, Institute of Liver Studies, King’s College Hospital, London SE5 9RS, UK; 2Department of Gastroenterology, King’s College Hospital, London SE5 9RS, UK; 3Department of Gastrointestinal Oncology, ENETS Centre of Excellence, Netherlands Cancer Institute, 1066 CX Amsterdam, The Netherlands; 4Department of Gastroenterology and Hepatology, Leiden University Medical Center, 2300 RC Leiden, The Netherlands; 5The ARDEN NET Centre, ENETS Centre of Excellence, University Hospitals Coventry & Warwckshire NHS Trust, Coventry CV2 2DX, UK

**Keywords:** GEP-NET, nutritional status, survival, sarcopenia

## Abstract

**Simple Summary:**

Neuroendocrine tumours are rare tumours arising in the digestive system, mainly the small bowel or pancreas. Due to their location in the digestive tract, NETs can cause symptoms of diarrhoea, abdominal pain or weight loss. These symptoms are often correlated with nutrition. Poor nutrition or malnutrition is well described. Sarcopenia is the loss of muscle mass or strength and a phenotype of malnutrition. In patients with cancer and sarcopenia, survival is poorer compared to patients without sarcopenia. There is little knowledge regarding sarcopenia and its effect on survival in patients with NETs. This study aims to describe the presence of sarcopenia at diagnosis of stage IV NET in the digestive system and correlate this with survival. Sarcopenia was present in 69% of patients. When there was a NET in the pancreas, the presence of sarcopenia was correlated with poorer survival.

**Abstract:**

Sarcopenia in patients with cancer is associated with adverse outcomes such as shorter survival. However, there exists little evidence regarding the prevalence of sarcopenia in patients with metastatic gastroenteropancreatic neuroendocrine tumours (GEP-NETs). Patients with a histologically confirmed newly diagnosed metastatic GEP-NET between 2006 and 2018, CT scan, and anthropometric data at diagnosis were included in this study. CT scans were analysed for the presence of sarcopenia and correlated with overall survival (OS). In total, 183 patients, 87 male (48%), with a median age of 62 years (IQR 52–68 years), were included. In 44 patients (24%), there was a pancreas NET, and in 136 patients, there was a small bowel NET (74%). Sarcopenia was present in 128 patients (69%) and unrelated to BMI (median 25.1). There were significant survival differences between patients with pancreatic and small bowel NETs at 86 vs. 141 months, respectively (*p* = 0.04). For patients with pancreatic NETs, the presence of sarcopenia was independently associated with shorter OS (HR 3.79 95% CI 1.1–13.03, *p*-value 0.035). A high prevalence of sarcopenia at the time of diagnosis of a metastatic GEP-NET was seen and associated with worse OS in patients with pancreatic NETs. Further research should focus on how to reverse sarcopenia and its impact on OS and/or quality of life.

## 1. Introduction

Neuroendocrine neoplasms (NENs) are rare neoplasms and consist of well-differentiated neuroendocrine tumours (NETs) and poorly differentiated neuroendocrine carcinoma (NECs). The incidence and prevalence of NENs are rising worldwide [1,2]. Diagnosing NEN can be a challenge due to the broad spectrum of symptoms. In some studies, up to 50% of patients reported symptoms for 2 to 5 years prior to diagnosis [3,4]. Due to the relative delay in diagnosis, metastatic disease is present in 25% of patients in population-based studies [1,2,5,6], but in tertiary centers, this figure is 60-80% [7,8,9,10]. Patients with NETs have a good prognosis even in metastatic disease, with the overall 5-year survival reaching 50–70% [2,5,6]. The prognosis for patients with a NET depends partly on histologic grading; grade 1 NET (G1, mitotic count less than 2/2 mm^2^ and Ki-67 less than 3%) has a better prognosis compared to grade 2 (G2, mitotic rate 2–20/2 mm^2^ and Ki-67:3–20%) or grade 3 (G3, mitotic rate greater than 20/2 mm^2^ and Ki-67 index greater than 20%) tumours [2]. Patients with NECs have a poor prognosis, with a median overall survival of 10 months [2]. Due to the differences in tumour morphology and overall survival, NETs and NECs should be seen as different entities. 

NETs arise from the enterochromaffin cells and can arise anywhere in the body but are mainly located in the lung or gastroenteropancreatic (GEP) tract, with the most common primary location being in the small bowel or pancreas [1]. The enterochromaffin cells can produce hormones resulting in functional NETs if the hormones reach the systemic circulation and cause symptoms [11].

Due to their position in the digestive tract and the long period of symptoms prior to diagnosis, patients with GEP-NETs are at risk of malnutrition. The prevalence of malnutrition, measured based on weight loss or low body mass index (BMI), is respectively 5–38% and 5–12% [12,13,14,15]. One study showed that patients with malnutrition had a prolonged hospital stay and poorer prognosis; however, this study included mainly patients with NECs [15]. 

Malnutrition is part of a spectrum of overlapping nutritional disorders, which includes underweight, disease-related cachexia and sarcopenia [16]. In 2019, the Global Leadership Into Malnutrition (GLIM) criteria for the diagnosis of malnutrition were published. In addition to weight, weight loss and BMI, the presence of sarcopenia is one of the criteria for malnutrition [17]. Sarcopenia is a muscle disease rooted in adverse muscle changes that occur across a lifetime; it is common among adults of older age but can also occur earlier in life [18]. In patients with cancer, sarcopenia is associated with an increased risk of chemotherapy toxicity, poorer outcomes of surgery, physical impairment and shorter survival [19]. The golden standard for diagnosing sarcopenia is body composition analysis on computer tomography (CT) or magnetic resonance imaging (MRI) scans [18]. Body composition can also diagnose adipopenia, which means fat mass depletion [20] and myosteatosis, which is fatty infiltration in the muscle and is used as a surrogate marker for muscle quality [21]. 

The prevalence of sarcopenia in patients with a recent diagnosis of gastrointestinal-located adenocarcinoma is 40–50% [19]. There are no studies reporting the prevalence of sarcopenia in patients with a recent diagnosis of a metastatic GEP-NET. It can be hypothesized that the prevalence is high due to the prolonged period of symptoms prior to diagnosis. One recent study included 104 patients with a recent diagnosis of GEP-NET, where 87% of patients had sarcopenia. However, only 52 patients had metastatic disease, and the prevalence of sarcopenia in this subgroup was not reported [22]. 

The aim of this study is to describe the prevalence of sarcopenia in patients with a recent diagnosis of metastatic GEP-NET and the association of sarcopenia with overall survival. 

## 2. Materials and Methods

### 2.1. Study Design and Study Population

This is a multi-centre retrospective study in 2 hospitals, the Netherlands Cancer Institute (NKI), Amsterdam, and the Netherlands and King’s College Hospital (KCH), London, the United Kingdom, both European Society of Neuroendocrine Tumours (ENETS) Centres of Excellence. The local prospective NET databases from both hospitals were screened for eligible patients. 

Inclusion criteria were adult patients (>18 years) with metastatic GEP-NETs diagnosed between 2006 and 2018 with a (low dose) CT scan of the abdomen at diagnosis. Based on this index CT scan, which was performed either at the referring hospital or at one of the study centres, the diagnosis of NET was made. A definite NET diagnosis was based on histopathological confirmation. Additional inclusion criteria were the availability of anthropometric data, including weight, height and body mass index (BMI). 

Exclusion criteria were stage I-III GEP-NET, missing CT scan of the abdomen at diagnosis, previous history of liver or kidney transplantation, second diagnosis of any other cancer within 1 year of follow-up, the presence of ascites or previous history of spinal surgery with metal implants. 

### 2.2. Outcomes

The primary outcome was the presence of sarcopenia at the time of diagnosis of a metastatic GEP-NET. The secondary outcome was the correlation of sarcopenia with overall survival. 

### 2.3. Body Composition Analysis

The body composition analysis was performed by two trained researchers (DC and SJ) according to previously published protocols [19,23]. A single slice of the CT scan at the lumbar level L3 was used, as this corresponds with total body muscle and fat mass [24,25]. Slice-O-Matic software (5.0 Rev-8, Tomovision, Milletta, QC, Canada) was used. Relevant tissues were identified based on their anatomical features and tagged with a colour as explained in Figure 1. The software multiplies preset Hounsefield units (HU) with pixels for the tagged area to calculate the relevant areas. The areas of interest, the thresholds and the definitions used are summarised in Table 1. Sarcopenia was present if the skeletal muscle area (SMA) corrected for height^2^ was below reference values. For myosteatosis as a surrogate marker for muscle quality, the muscle attenuation in Hounsefield units was used; myosteatosis was considered to be present if it was below the reference value. The subcutaneous, visceral and intermuscular adipose tissue areas were added up to calculate the total adipose tissue area. If this area was below the reference value, adipopenia was present. 

### 2.4. Additional Data Collection

Baseline demographic characteristics were collected, including histology details for grading and staging according to the WHO 2019 classification [27]. TumorNodesMetastasis (TNM) staging was based on the *Union for International Cancer Control (UICC) 8th Edition* (2016). A functional tumour was diagnosed if there was diarrhoea and/or flushing in the presence of raised 5-hydroxyindolecactic Acid (5-HIAA) in the urine or raised serotonin levels in the blood [11]. The World Health Organisation (WHO) performance status classification was used for vitality. The Charlson comorbidity index was used for comorbidity scoring [28]. Body mass index (BMI) was calculated. Underweight was defined as BMI < 18.50 kg/m^2^, normal weight as 18.51–24.99 kg/m^2^, overweight as 25.00–29.99 kg/m^2^ and obese as > 30 kg/m^2^ [23]. Patients were observed until death or 1 July 2021, at which time they were censored. Overall survival was calculated between the date of diagnosis and the date of death or censored end date. 

### 2.5. Statistical Analysis

Data were analysed using SPSS version 27 (IBM, NY, USA). Data were displayed as median with interquartile range. Student’s *t*-tests, Mann–Whitney U tests and chi-squared tests were performed to describe differences between groups and results. For the survival analysis, Kaplan-Meier curves were plotted, and the log-rank test was performed. Cox regression analysis was performed to correct for confounders, including age, sex, and grading and to explore the presence of sarcopenia as an independent risk factor for overall survival. A *p*-value of <0.05 was considered significant.

### 2.6. Ethical Approval 

The study was conducted in accordance with the Declaration of Helsinki and the protocol was approved by the Institutional Review Board (or Ethics Committee) of the Netherlands Cancer Institute (IRBd19–065, 2–6–2019) and the Health Research Authority of the United Kingdom (IRAS number 246990 24–9–2018). 

## 3. Results

### 3.1. Patient Demographics

A total of 183 patients were included, of which 48% were male, with a median age of 62 years. The baseline characteristics are summarized in Table 2. The main locations for metastases were the liver (89%), lymph nodes (40%), mesentery (27%), peritoneum (11%), bone (10%) or other (18%). 

### 3.2. Body Composition and Nutritional Markers

The median BMI in this cohort was 25 (22.6–29.4), with 52% of patients being overweight or obese. Patients with a grade 2 NET had a significantly lower BMI than patients with a grade 1 NET (24 vs. 25.8, *p*-value 0.02). 

Sarcopenia was present in 126 patients (69%). Patients with a grade 1 (68%) and 2 (70%) NET were equally distributed among those with sarcopenia (*p*-value 0.097). 

The dispersion of sarcopenia differed significantly between BMI categories, as follows: patients who were underweight, *n* = 7 (78%); normal weight, *n* = 57 (74%); overweight, *n* = 42 (74%); and obese, *n* = 19 (51%) (*p*-value 0.045). Patients with sarcopenia had poorer muscle quality compared to patients without sarcopenia. Patients with sarcopenia had a significantly lower visceral adipose tissue area (*p*-value 0.04), but the presence of adipopenia did not differ between them. Details are summarized in Table 2. 

### 3.3. Sarcopenia Per Location of The Primary Tumour

There were 44 patients with a pancreatic NET and 136 patients with a small bowel NET (Table 3). Patients with pancreatic NETs had mainly G2 tumours, and patients with small bowel NETs had mainly G1 tumours (*p*-value < 0.001). The primary location of the tumour did not influence the presence of sarcopenia. The only difference in body composition between patients with pancreatic and small bowel NETs was the muscle quality, which was significantly more frequently poor in patients with small bowel NETs (*n* = 63, 46%) vs. pancreatic NETs (*n* = 10, 23%) (*p*-value 0.006). 

### 3.4. Survival Analysis

The median survival was 119 months (IQR 84–153 months), and the 5-year overall survival was 75% (Figure 2A). The types of treatment for NETs during the follow-up are described in Table 4. Patients with pancreatic NETs underwent significantly less surgical treatment compared to patients with small bowel NETs (*p*-value < 0.001). Chemotherapy and mammalian target of rapamycin (mTOR) or tyrosine kinase inhibitors (TKI) were significantly more often used in patients with pancreatic NETs (*p*-value *p* < 0.001). 

The median survival for patients with a pancreatic NET was significantly shorter than for patients with a small bowel NET (86 months (IQR 31–140 months) vs. 141 months (IQR 108–173 months), *p*-value < 0.001) (Figure 2B). For pancreatic and small bowel G1 tumours, the median survival was not reached. The median survival for G2 tumours differed significantly between pancreas and small bowel tumours at 67 months vs. 100 months, *p*-value < 0.001. 

There are no significant differences (*p*-value 0.39) in overall survival between patients with and without sarcopenia in the whole cohort (Figure 2C). For patients with small bowel NETs, the presence or absence of sarcopenia does not influence the overall survival (*p*-value 0.49), Figure 2D. In patients with pancreatic NETs, there was a significant (*p*-value 0.045) survival difference between patients without sarcopenia (median not reached) and patients with sarcopenia (median 50 months, IQR 31–140 months), as seen in Figure 2E. 

Uni- and multivariate analyses were performed separately per location of the primary tumour, as displayed in Table 5. Sarcopenia was significantly associated with survival in patients with pancreatic NETs (HR 3.79, 95% CI 1.1–13.03) after correction for confounders (Table 5). Age (HR 1.05 95% CI 1.01–1.09) and grading (HR 4.5 95% CI 2.26–9.15) were significantly associated with small bowel NET survival (Table 6). 

## 4. Discussion

This is the largest study regarding sarcopenia in patients with a stage IV GEP-NET, and it suggests a high prevalence of sarcopenia. The reported prevalence of sarcopenia of 70% at the time of diagnosis of metastatic gastrointestinal NET is higher than in patients with metastatic adenocarcinoma (30–60%) [29,30,31]. Studies regarding patients with metastatic pancreatic adenocarcinoma reported a prevalence of sarcopenia of 56–86% at diagnosis [31,32,33]. Although pancreatic adenocarcinoma behaves differently and has a poor prognosis compared to pancreatic NETs, the prevalence of sarcopenia is comparable. In those studies, the presence of sarcopenia was not associated with overall survival in uni- and multivariate analyses [33,34].

The 72% prevalence of sarcopenia in patients with metastatic small bowel NETs is much higher than the 27–44% prevalence in patients with metastatic colorectal cancer [29,31]. There are no studies evaluating sarcopenia in small bowel adenocarcinoma. The higher prevalence of sarcopenia in patients with metastatic GEP-NETs can be explained by a longer period of symptoms, delay in diagnosis, and possible malabsorption [3,4]. It is difficult to compare the findings in this study with the only two available studies regarding sarcopenia in patients with GEP-NETs, as both used different study populations [22,35]. 

Neither the location of the primary tumour nor the grading of the NET predicts the prevalence of sarcopenia. This has not been studied previously, as other studies regarding sarcopenia in patients with NET did not run separate analyses per primary tumour or grading [22,35]. 

This study highlights the high prevalence of sarcopenia in patients with a normal BMI (42%) or BMI > 25 kg/m^2^ (52%). Assessing a patient’s nutritional status based only on BMI will result in missing sarcopenia. Other studies regarding patients with metastatic pancreatic or colorectal adenocarcinoma report similar findings [32,33,34,36]. With the increase in obesity in the general population, assessing sarcopenia becomes more important, as BMI alone will not provide adequate information [37].

The median BMI did not differ between patients with pancreatic or small bowel NETs. However, it could be suggested that patients with pancreatic NETs have more frequent pancreatic exocrine insufficiency (PEI), leading to steatorrhoea, malnutrition and, subsequently, sarcopenia. A recent publication on patients with pancreatic NETs reported improved survival in patients with PEI and the use of pancreatic enzyme replacement therapy (PERT), although this was a small cohort, and PERT was not independently associated with survival [38]. 

The 5-year overall survival was 75% in this study, which is in line with the 50–70% 5-year OS reported in the literature [2,6,39]. However, OS differed significantly between patients with pancreatic NETs (5-year OS 53%) and small bowel NETs (5-year OS 81%). These differences have been described previously, for example, in the USA Surveillance, Epidemiology and End Results (SEER) database [2]. Differences in overall survival between pancreatic and small bowel NETs in this study could also partly be explained due to differences in methods of grading. The pancreatic NETs had significantly more grade 2 tumours, while the small bowel NETs were more often grade 1 tumours. The differences in overall survival between grade 1 and grade 2 tumours were described in previous studies [2,8]. This study demonstrates a median overall survival of 86 months for patients with metastatic small bowel NETs; due to the chosen study period and censoring of patients, the number of patients in follow-up after 120 months becomes very low. At 132 months, there were only two patients in the study. However, one died, and at 144 months, there was only 1 patient left. This resulted in a drop in the survival curve from 80% to 40% at this point. 

In the entire cohort, there is no survival difference between patients with and without sarcopenia. This could be explained due to the predominance of small bowel NETs compared to pancreatic NETs, which showed a significant survival difference between sarcopenia and no sarcopenia and was confirmed in multivariate analysis. This effect has never been described before in the literature. The only comparable study regarding patients with all stages of GEP-NET showed that patients with sarcopenia have a non-significant (*p*-value 0.12) worse overall survival compared to patients without sarcopenia. There was no correction for confounders in this study [22]. 

This study highlights the importance of good nutritional status in patients with metastatic GEP-NETs, especially in patients with pancreatic NETs. Screening for malnutrition and treatment with PERT should be initiated from diagnosis as it has been shown to improve survival in patients with pancreatic adenocarcinoma [40]. Patients with metastatic GEP-NET and avidity on somatostatin receptor imaging will be treated with a somatostatin analogue (SSA), which can cause pancreatic exocrine insufficiency [41,42]. Further prospective research should include the role of PERT and its effect on the reversibility of sarcopenia. It may be helpful to refer most NET cases for a nutrition assessment early in the course of the disease.

Early diagnosis of malnutrition and sarcopenia could be treated and could possibly improve survival. Future research should focus on developing easy-to-use tools to identify patients at risk of malnutrition and sarcopenia early in the course, as the current golden standard is muscle mass analysis at the lumbar level L3 on cross-sectional imaging, which might not be available in every center. Research should also focus on treatment to reverse sarcopenia and its effect on survival, the reduction of gastrointestinal-related symptoms and improvements in quality of life. Recent studies in patients with incurable lung or gastrointestinal cancer demonstrated a lower quality of life in patients with sarcopenia compared to patients with sarcopenia [43,44]. 

This study has some limitations. The first one is bias, as patients were selected from large tertiary referral centres, and the inclusion criteria regarding the presence of CT-scan and anthropometric data were strict. Therefore, patients could be missed, and findings could not be extrapolated to smaller or regional NET centres. Another limitation of this study is the lack of correlation with weight loss. 

## 5. Conclusions

Sarcopenia is frequently present in patients with a recent diagnosis of metastatic GEP-NET, irrespective of BMI. In patients with pancreatic NETs, sarcopenia is related to worse overall survival.

## Figures and Tables

**Figure 1 cancers-15-00782-f001:**
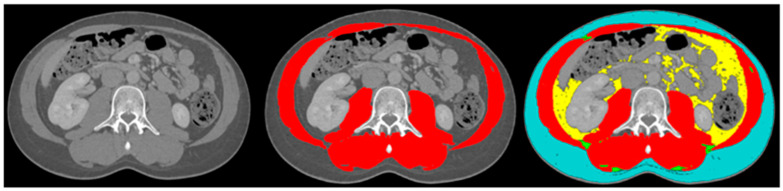
Example body composition analyses. Left image: cross-sectional image at lumbar level L3; middle image: tagged muscle mass (red colour); right image: tagged subcutaneous adipose tissue (blue colour), visceral adipose tissue (yellow colour), intermuscular adipose tissue (green colour) and muscle mass (red colour).

**Figure 2 cancers-15-00782-f002:**
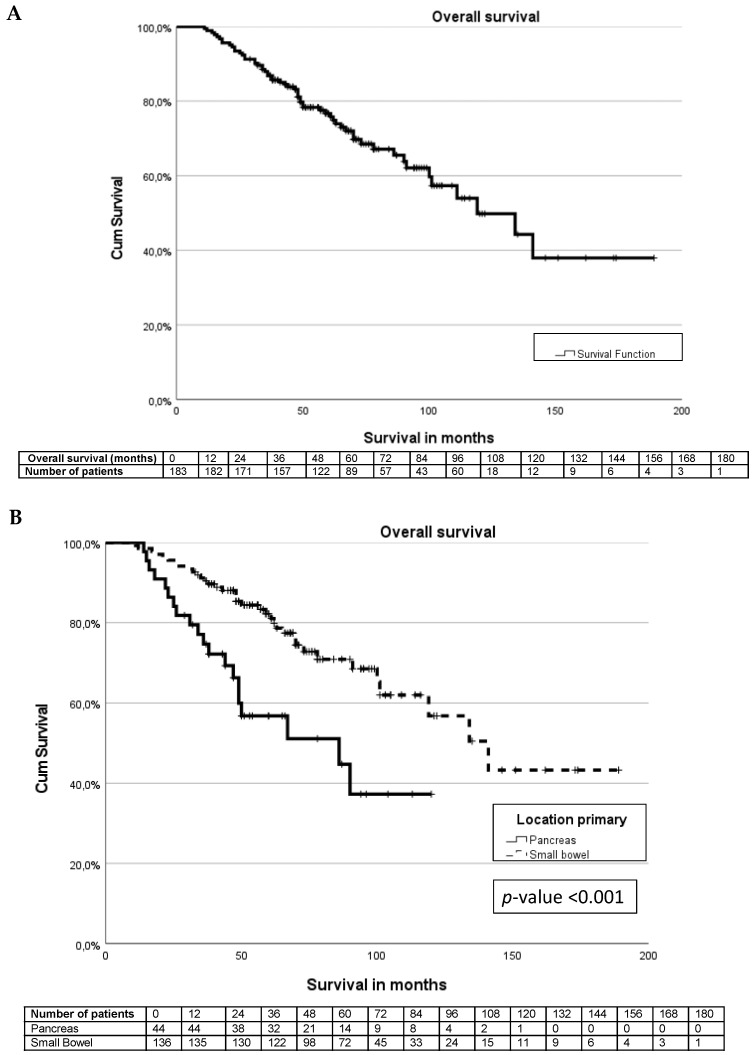
Kaplan–Meier curves. (**A**) Survival curve for entire cohort. (**B**) Survival differences between pancreas and small bowel NET. (**C**) Survival differences between sarcopenia and non-sarcopenia. (**D**) Survival differences between small bowel NET sarcopenia and non-sarcopenia. (**E**) Survival differences between pancreas NET with sarcopenia and non-sarcopenia.

**Table 1 cancers-15-00782-t001:** Body composition analysis.

Area to Analyse	Area’s	Settings	Diagnosis	Refence Value Male	Reference Value Female
**Muscle**	Skeletal muscle area (SMA) (cm^2^)	−29 HU to +150 HU			
	Skeletal muscle index (SMI) (cm^2^/m^2^)	SMA corrected for height^2^	**Sarcopenia** [23]	If BMI > 25 SMI > 53 cm^2^/m^2^	>41 cm^2^/m^2^
				If BMI < 25 SMI > 43 cm^2^/m^2^	
**Muscle quality**	All muscle area	−29 HU to +150 HU	**Myosteatosis** [23]	If BMI > 25 MQ > 33 HU	If BMI > 25 MQ > 33 HU
				If BMI < 25 MQ > 41 HU	If BMI < 25 MQ > 41 HU
**Adipose tissue**	Subcutaneous adipose tissue (SAT) (cm^2^)	−150 HU to −50 HU		>132.3 cm^2^	>261.8 cm^2^
	Visceral adipose tissue (VAT) (cm^2^)	−190 HU to −30 HU		>160 cm^2^	>80 cm^2^
	Intermuscular adipose tissue (IMAT) (cm^2^)	−190 HU to −30 HU		12–14 cm^2^	12–14 cm^2^
	Total adipose tissue (TAT) (cm^2^)	Sum SAT + VAT + IMAT	**Adipopenia** [26]	>364 cm^2^	>318 cm^2^

**Table 2 cancers-15-00782-t002:** Body composition analysis and baseline characteristics.

	All Patients *n* = 183	Sarcopenia *n* = 126	Non-Sarcopenia *n* = 57	*p*-Value
**Median BMI**, median, (IQR)	25.1 (22.6–29.4)	24.7 (22.2–27.8)	26.4 (23.7–31)	**0.013**
**BMI categories**				**0.045**
1. Underweight, n (%)	9 (5)	7 (6)	2 (4)	
2. Normal weight, n (%)	77 (42)	57 (45)	20 (35)	
3. Overweight, n (%)	57 (31)	42 (33)	15 (26)	
4. Obese, n (%)	39 (21)	19 (15)	20 (35)	
**Muscle attenuation**	39.1 (32.5–45.4)	38.9 (30.5–45.4)	40.6 (35.7–47.7)	0.072
**Myosteatosis**, n (%)	75 (41)	60 (48)	15 (26)	**0.007**
Subcutaneous adipose tissue	154.9 (97.1–231.5)	146.7 (92.8–223.6)	157 (94.5–223.6)	0.85
Visceral adipose tissue	95.1 (35.8–179.6)	79.6 (23.5–144.5)	136.8 (60.8–180.5)	**0.04**
Intermuscular adipose tissue	6.3 (3.5–10.9)	6.7 (3.4–11.3)	5.1 (3–9.3)	0.17
Total adipose tissue	285.1 (157.7–431.6)	260.9 (142.4–371.5)	308.6 (146–433.9)	0.22
**Adipopenia**, n (%)	112 (61)	80 (63)	32 (56)	0.79
**Sex**				
Male n (%)	87 (48)	55 (44)	32 (56)	0.117
Female n (%)	96 (52)	71 (56)	25 (44)	
**Age at diagnosis**, median (IQR)	62 (52–68)	60.9 (53.8–69)	59.2 (50.5–67)	0.286
**Ethnicity**				
White, n (%)	153 (85)	102 (81)	51 (89)	0.984
BAME, n (%)	16 (9)	12 (9.5)	4 (7)	
Unknown, n (%)	14 (8)	12 (9.5)	2 (4)	
**WHO status**				
0–1, n (%)	129 (71)	85 (67)	44 (77)	0.562
2, n (%)	34 (19)	11 (9)	23 (40)	0.403
**Charlson comorbidity score**, median (IQR)	2 (1–3)	1.8 (1–3)	1.6 (1–3)	0.61
**Primary NET location**				0.271
Pancreas, n (%)	44 (24)	26 (21)	18 (32)	
Small bowel, n (%)	136 (74)	98 (78)	38 (67)	
Other, n (%)	3 (2)	2 (1.6)	1 (1.8)	
**Grading**				0.949
G1, n (%)	107 (59%)	73 (58%)	34 (60%)	
G2, n (%)	71 (39%)	50 (40%)	21 (37%)	
G3, n (%)	3 (2%)	2 (1.6%)	1 (1.8%)	
**Functioning**, n (%)	96 (52%)	70 (56%)	56 (98%)	0.387

BMI: body mass index; BAME: Black, Asian and minority ethnicity; WHO: World Health Organization; G1: grade 1; G2: grade 2; G3: grade 3.

**Table 3 cancers-15-00782-t003:** Differences between patients with pancreatic and small bowel NET.

	Pancreas (*n* = 44)	Small Bowel (*n* = 136)	*p*-Value
**Sex**: Male	*n* = 22 (50%)	*n* = 65 (47.8%)	0.06
**Median age at diagnosis**	56.5 (47.3–67.0)	63.0 (55.0–69.0)	**0.003**
**Ethnicity**: White	*n* = 38 (86.4%)	*n* = 112 (82.4%)	0.54
**WHO status**			0.2
0–1	*n* = 33 (75%)	*n* = 96 (70.6%)	
2	*n* = 5 (11%)	*n* = 28 (20.6%)	
**Charlson comorbidity score**	1.0 (1.0–3.0)	2.0 (1.0–3.0)	0.13
**Grading**			**<0.001**
G1	*n* = 15 (34.1%)	*n* = 90 (66.2%)	
G2	*n* = 26 (59.1%)	*n* = 44 (32.4%)	
G3	*n* = 2 (4.5%)	*n* = 1 (0.7%)	
**Functioning**	*n* = 10 (22.7%)	*n* = 85 (62.5%)	**<0.001**
**Median BMI**	24.8 (22.7–27.6)	25.2 (22.4–29.4)	0.92
**BMI category**			0.29
1 Underweight	*n* = 2 (4.5%)	*n* = 7 (5.1%)	
2 Normal weight	*n* = 20 (45.5%)	*n* = 56 (41.2%)	
3 Overweight	*n* = 12 (27.3%)	*n* = 45 (33.1%)	
4 Obese	*n* = 10 (22.7%)	*n* = 28 (20.6%)	
**Sarcopenia**	*n* = 26 (59%)	*n* = 98 (72%)	0.11
**Myosteatosis**	*n* = 10 (23%)	*n* = 63 (46%)	**0.006**
**Adipopenia**	*n* = 32 (73%)	*n* = 78 (57%)	0.10

WHO performance status: World Health Organization performance status, BMI: body mass index, G1: grade 1, G2: grade 2, G3: grade 3.

**Table 4 cancers-15-00782-t004:** Treatment details.

Treatment Details	Pancreas (*n* = 44)	Small Bowel (*n* = 136)	*p*-Value
Surgery, *n* (%)	13 (30)	92 (68)	**<0.001**
SSA, *n* (%)	37 (77)	120 (88)	0.47
CTx/mTOR/TKI, *n* (%)	32 (73)	9 (7)	**<0.001**
PRRT, *n* (%)	17 (39)	44 (32)	0.44
Embolisation, *n* (%)	4 (9)	22 (16)	0.25
Liver-directed therapy, *n* (%)	0	6 (4)	0.16

SSA: somatostatin analogue, CTx chemotherapy, mTOR: mammalian target of rapamycin, TKI: tyrosinase kinase inhibitor (TKI).

**Table 5 cancers-15-00782-t005:** Uni- and multivariate analyses for pancreas NET.

Pancreas NET	Univariable Analysis		Multivariable Analysis	
Variables	HR (95% CI)	*p*-Value	HR (95% CI)	*p*-Value
Age at diagnosis	0.98 (0.95–1.02)	0.39	0.98 (0.94–1.02)	0.40
Sex	0.68 (0.32–2.16)	0.7	0.45 (0.14–1.50)	0.16
BMI	0.97 (0.86–1.09)	0.59		
Sarcopenia	3.01 (0.97–9.34)	**0.057**	3.79 (1.1–13.03)	**0.035**
Grading	1.25 (0.46–3.42)	0.66	2.08 (0.61–7.03)	0.24

**Table 6 cancers-15-00782-t006:** Uni- and multivariate analyses for small bowel NET.

Small Bowel NET	Univariable Analysis		Multivariable Analysis	
Variables	HR (95% CI)	*p*-Value	HR (95% CI)	*p*-Value
Age at diagnosis	1.05 (1.01–1.09)	**0.01**	1.05 (1.01–1.09)	**0.01**
Sex	0.80 (0.41–1.55)	0.51	0.94 (0.48–1.83)	0.85
BMI	0.98 (0.91–1.05)	0.5		
Sarcopenia	1.32 (0.6–2.89)	0.31	0.83 (0.36–1.91)	0.66
Grading	4.51 (2.30–8.81)	**<0.001**	4.5 (2.26–9.15)	**<0.001**

## Data Availability

All data generated during this study are included in this article. Further enquiries can be directed to the corresponding author.

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
