# Peer review of "Prevalence of Sarcopenia and Impact on Survival in Patients with Metastatic Gastroenteropancreatic Neuroendocrine Tumours"

_cancers, 2023, doi:10.3390/cancers15030782_

Round 1

Reviewer 1 Report

This is an interesting study providing novel and useful insight in the importance of sarcopenia in NEN patients. A minor comment/suggestion: Paragraph in lines 317-320 seems to be out of place (general instructions to authors?) 2.

Author Response

This is an interesting study providing novel and useful insight in the importance of sarcopenia in NEN patients.

We would like to thank the reviewer for this nice feedback.

A minor comment/suggestion: Paragraph in lines 317-320 seems to be out of place (general instructions to authors?) 2.

Thank you for this comment, these standard lines were removed.

Reviewer 2 Report

The paper "prevalence of sarcopenia and impact on survival in patients with metastatic GEP-NETs" is a retrospective bicenter study reporting the prevalence and prognostic significance of sarcopenia in a cohort of 183 patients. It is a sound and well written study that could be of interest to the readers of Cancers. However, there are some minor points that could potentially improve its comprehension.

Title: It is not advisable to use acronyms without explanation in the title, in order not to discourage not so specialized readers and therefore not so used to them. 

Introduction: It is tidy, well written and relevant. However, some style corrections could be useful. For example, some phrases are like two expressions united by a comma and could be shortened or split in, as the ones in lines 48-50 and 70-72.

Study design: Did the authors register the protocol before the study? Was the study protocol approved by the ethics committee at the authors' centers? If so, they should give  the references. If not, they should explicitly include this as a limitation. Abscence of pre-study register could favour changing its design depending on findings and data fishing bias.

Statistical analysis: I do not understand including the presence or absence of sarcopenia as a correcting confounder in multivariate analysis when it is the primary outcome of the study. Please explain.

Tables: 
- The units in which the variables are given, such as "n (%)" and "median (IQR)" should be only in the left column, after the title of the variable. 
- Tables 2 and 3 should be united in one table, as in table 4. 
- In gender and ethnicity lines, reporting only male and white is not advisable.

Figures: 
- Figure 1A is redundant and could be avoided. 
- I suggest using a clearer differentiation between both lines in the Kaplan-Meier graph, in order to allow comprehension in black and white printings.

Results: 
- Line 208: "regardless location of the primary tumour or grading" should be changed into "in the whole cohort".
- Lines 214-216: The phrase "Due to the significant (...) in table 5" is not to be included in the results section. If the decision of making different survival analysis for pancreas and intestinal tumours was decided before the study, it should be reflected in the methods section. Otherwise the phrase should be deleted, as it could be regarded as a data fishing bias.

Discussion: 
- Some conclusions along the discussion are far too categorical for a small cohort study: "demonstrates that 70%..." (line 223) should be changed into "suggests a high prevalence...". 
- The findings of the study should be given one by one in separated paragraphs followed by their specific discussion, instead of all together in the first paragraph.
- Discussion is too long and would benefit from shortening. For example, a paragraph from lines 241 to 247 is redundant and could be deleted. A paragraph from lines 258 to 270 could be summarized in two short sentences explaining the global findings of all cited studies. A paragraph from lines 297 to 308 could also be summarized in a few short sentences.
- The phrase in line 309 ("This is the largest...") should be placed at the beginning of the discussion section, toguether with the main finding of the study. The two following sentences ("The cohort is (...) are available.") could be avoided. 
- The limitations paragraph should be separated and clearly stated as such, beggining with a standard formula, such as "This study has some limitations, being the main one..."

Author Response

We would like to thank the reviewer for the nice feedback and suggestions how to improve the manuscript.

We have answered the reviewer’s comments in full and made the appropriate changes as outlined. In the attached file is the response to each individual point. We hope that we have addressed these issues adequately. 

Reviewer 3 Report

Clement and colleagues sought to examine the prevalence of sarcopenia in patients with metastatic gastroenteropancreatic neuroendocrine tumours. This question is relevant and important; however, there are some concerns that need to be addressed prior to publication. Most importantly, more details in the methodology are needed and the introduction needs to be revamped to better introduce the variables/terms rather than a literature review of the disease conditions.   

The introduction read more like a review of the literature rather than an introduction to the paper and variables. Consider summarizing/consolidating the larger epidemiological studies and disease characteristics/definitions aimed at shortening the introduction length.

Line 75-81: The loss of muscle mass and quality secondary to chronic disease (like cancer) is better termed “cachexia” rather than sarcopenia. The authors correctly state that sarcopenia is more commonly used to define the loss of muscle with aging; however, the overlap between sarcopenia and cancer has been more typically linked to the aged population of cancer patients – the average age of a cancer diagnosis is ~65. Therefore, sarcopenia is linked to cancer mortality and morbidity as a precondition prior to cancer diagnosis. The loss of mass with cancer is better defined as cachexia. It is difficult to distinguish between the two conditions clinically as the onset of the malignancy does not occur upon diagnosis as it is possible that the earlier presence of the cancer is what is driving the weight loss. While this may appear trivial, sarcopenia and cachexia have differing etiologies and mechanisms of action and conflating the 2 does not benefit the medical and scientific communities. The authors do not give this sufficient attention and therefore are missing a large portion of the literature and nuance of the disease paradigm. It is recommended that the authors either address the overlap of sarcopenia and cachexia, or better define the outcomes/variables in the introduction and delimit the study to investigating the presence of sarcopenia at diagnosis (as is already indicated in your methods).

How many instances was "last follow-up appointment" used as the terminal metric? How can the reader be certain that "last follow-up appointment" means that a death occurred? 

It would be beneficial to switch table 2 and table 3. Since you are dichotomizing the populations, it would benefit the reader to understand the dichotomy prior to all other disease metrics.

It is not clear how “muscle quality” is defined and characterized. There are no listed metrics of muscle function and obtaining “quality” purely from scans requires more detail. This should be interpreted with extreme caution.

How is adipopenia defined and characterized?

Representative images from the scan would be helpful for the reader.

Table 4 and lines 173-177 will confuse the reader as the methods proclaim the outcome measures were obtained at diagnosis. Introduction of patient treatments should be shown with survival rather than sarcopenia. From the writing, it is my impression that “treatment details” would not have impacted sarcopenia prevalence as these patients had not been treated yet. Rather “treatment details” would impact survival. I recommend making treatment details its own table.

Crop figure 1E’s top border as it is overlapping with Figure 1C and Figure 1D.

Figure 1 needs are figure legend. It is not apparent what “no at risk” means.

Line 224-225 needs more explanation. It is not evident what is meant when the authors say “BMI does not predict” followed by, “is more prevalent in low BMI”. The second clause indicates to me that having a low BMI would “predict” a higher likelihood of having sarcopenia.

The drop from 80% to 40% survival in Figure 1D requires significant discussion. This data appears skewed given the low n of “no sarcopenia” compared to “sarcopenia” in small bowel NETs.

Author Response

We would like to thank the reviewer for this nice and worthwhile feedback. We made changes as required summarized per point in the attached file. We hope that we have addressed these issues adequately. 

Round 2

Reviewer 2 Report

I congratulate the authors for this prompt, extensive and rigorous revision of the manuscript. I do not have any further comments.